# Winter habitats of bats in Texas

Melissa B. Meierhofer[1,2]*, Joseph S. Johnson[3], Samantha J. Leivers[2], Brian L. Pierce[2], Jonah E. Evans[4], Michael L. Morrison[1]

**1** Department of Wildlife and Fisheries Sciences, Texas A&M University, Texas, United States of America, **2** Natural Resources Institute, Texas A&M University, Texas, United States of America, **3** Department of Biological Sciences, Ohio University, Athens, Ohio, United States of America, **4** Wildlife Diversity Program, Texas Parks and Wildlife Department, Boerne, Texas, United States of America

* melissa.meierhofer@ag.tamu.edu

**Data Availability Statement:** All data files are available from the Texas Data Repository: https://doi.org/10.18738/T8/BDP6XO.

**Funding:** This work was funded by the U.S. Fish and Wildlife Service's State Wildlife Grant Program

## Abstract

Few studies have described winter microclimate selection by bats in the southern United States. This is of particular importance as the cold-adapted fungus, *Pseudogymnoascus destructans*, which causes the fatal bat disease white-nose syndrome (WNS), continues to spread into southern United States. To better understand the suitability of winter bat habitats for the growth of *P. destructans* in this region, we collected roost temperature and vapor pressure deficit from 97 hibernacula in six ecoregions in Texas during winter 2016–17 and 2017–18. We also measured skin temperature of Rafinesque's big-eared bats (*Corynorhinus townsendii*), Townsend's big-eared bats (*C. townsendii*), big-brown bats (*Eptesicus fuscus*), southeastern myotis (*Myotis austroriparius*), cave myotis (*M. velifer*), tri-colored bats (*Perimyotis subflavus*), and Mexican free-tailed bats (*Tadarida brasiliensis*) during hibernation to study their use of torpor in these habitats. We found that temperatures within hibernacula were strongly correlated with external air temperatures and were often within the optimal range of temperatures for *P. destructans* growth. Hibernacula and skin temperatures differed among species, with Rafinesque's big-eared bats, southeastern myotis, and Mexican free-tailed bats occupying warmer microclimates and having higher torpid skin temperatures. For species that were broadly distributed throughout Texas, hibernacula and skin temperatures differed within species by ecoregion; Tri-colored bats and cave myotis in colder, northern regions occupied colder microclimates within hibernacula and exhibited colder skin temperatures, than individuals of the same species in warmer, southern regions. These data illustrate the variability in microclimates used as hibernacula by bats in Texas and suggest similar variation in susceptibility to WNS in the state. Thus, monitoring microclimates at winter roosts may help predict where WNS may develop, and where management efforts would be most effective.

## Introduction

The choice of overwintering habitat has numerous impacts on mammalian hibernators. Suitable hibernacula must provide an appropriate thermal environment. Defining appropriate thermal habitats has been the focus of research for decades, as the temperatures inside

(CFDA# 15.611) as administered by Texas Parks and Wildlife Department, https://tpwd.texas.gov/ and the U.S. Fish and Wildlife Service (CFDA# 15.657), https://www.fws.gov/ (MLM, BLP). Additional funding support was provided by the Fightwns 'Micro Grants for Microbats', https://www.fightwns.org/, Texas Ecolab research grants, https://texasecolab.org/, and National Speleological Society WNS Rapid Response Grant, https://caves.org/WNS/Rapid_Response.shtml (BLP, MBM). The funders had no role in study design, data collection and analysis, decision to publish, or preparation of the manuscript.

**Competing interests:** The authors have declared that no competing interests exist.

hibernacula profoundly influence the metabolic rates, and, therefore, the energy budgets of mammalian hibernators [1–3]. During torpor, metabolic rates decline with ambient temperatures until conditions fall below a hibernator's thermoregulatory set point, at which point metabolic rate increases [1]. Although this relationship indicates that there are temperatures where energetic savings are greatest, hibernators often select greater temperatures for both energetic and non-energetic reasons [4–6]. Thus, habitat selection is central to the winter ecology and physiology for mammalian hibernators, and an understanding of this process can help predict how species will be affected by continued environmental change [7].

Many studies of North American subterranean bat species are focused on regions with cold winter climates (e.g., [8–9]), with less research conducted in warmer, more southern regions of the continent [10–12]. However, research outside of North America shows that seasonal torpor is not limited to cold climates or areas with severe winters. Hibernation in warmer climates appears to be typified by short bouts of torpor and relatively high torpid body temperatures [13]. Although energetic challenges of winter in these warm regions may seem less compelling than the challenges posed by colder climates, the use of seasonal torpor in subtropical regions provides testament to its importance in the life cycles of many mammals [13–17].

Unfortunately, microclimates suitable for hibernation may also place bat species in North America at risk to white-nose syndrome (WNS) [18–20]. WNS is an emerging infectious disease of North American bats caused by the cold-tolerant fungus, *Pseudogymnoascus destructans*, which thrives in caves, underground mines, and other features that many bats use as hibernacula [18, 21–22]. *P. destructans* infects dermal tissue of hibernating bats, leading to dehydration and death [23–25]. The fungus has spread rapidly from where it was first documented in North America in New York State in 2006, annually spreading to new regions [18, 26–27]. *P. destructans* grows optimally between 12.5˚ and 15.8˚ C, although it can persist at temperatures from 3.0–19˚ C [20]. Thus, *P. destructans* can survive in a range of habitats that are important for bats during hibernation and is likely to continue to spread to new regions and cause mass mortalities [28].

Because the effects of WNS vary among species and environmental temperatures [19, 29–30], a detailed understanding of winter habitats used by different species in different regions is needed as the causative fungus continues to spread. The need for these data is illustrated by the recent spread of *P. destructans* into southern United States, including Texas [31–33] where it is still uncertain if bats will develop WNS, and if so, how severe mortality might be. Although Texas bats first tested positive for *P. destructans* during spring 2017, no WNS mortalities or bats with histopathological symptoms of the disease have been confirmed in the state [31–33]. Understanding the unique winter ecologies of WNS-susceptible species in the southern United States is therefore imperative in understanding the spread of *P. destructans* and determining species and regions most at risk from WNS.

Based on existing studies, it would be reasonable to predict notably different species-specific outcomes as WNS expands to hibernacula located in warmer climates such as Texas. Webb and colleagues [8] summarized inter- and intraspecies variations in the temperatures at which various bat species were found hibernating. Although most of the data reported were collected from northern geographic ranges of North America, the authors suggested that species inhabiting cold climates can hibernate in colder conditions than species inhabiting warm climates. This prediction is notable because warmer hibernacula temperatures would likely result in higher torpid body temperatures, higher torpid metabolic rates, and greater rates of periodic arousals from torpor [34–36]. If bats in subtropical regions are adapted to frequent arousals from winter torpor, such as those seen in WNS-affected bats, impacts from the disease may be less than observed in the temperate region of North America. However, the higher hibernacula

temperatures that one might expect in the subtropics may result in more favorable conditions for *P. destructans* to thrive compared to those farther north [11, 20], potentially resulting in more severe infections. Thus, WNS pathology in areas such as Texas is difficult to predict in the absence of data on winter habitat selection and behavior in the region. Despite this need, few researchers have studied hibernation in southern populations of bat species with broad distributional ranges [11, 37–39].

To improve our overall understanding of the suitability of bat habitats for *P. destructans*—and potentially WNS—in Texas, and to better inform management practices, we described the microclimates used by seven bat species as hibernacula across the state. We predicted that temperatures inside hibernacula would strongly correlate with external temperatures, and would be suitable for *P. destructans* growth. Based on species distribution and previous research on winter activity [40–42], we predicted that Rafinesque's big-eared bats (*Corynorhinus rafinesquii*), southeastern myotis (*Myotis austroriparius*), and Mexican free-tailed bats (*Tadarida brasiliensis*) would select microclimate within hibernacula having higher ambient temperatures and lower vapor pressure deficit (VPD) than all other species, and would subsequently have warmer torpid skin temperatures. We hypothesized that two species commonly found throughout Texas, the cave myotis (*Myotis velifer*) and tri-colored bat (*Perimyotis subflavus*), would exhibit variation in their microclimate selection across the state. Because bats hibernating in cold climates would be exposed to lower ambient roost temperatures, we predicted that they would therefore have lower skin temperatures than bats in warm climates. Finally, we predicted that tri-colored bats hibernating in culverts would have higher skin temperatures than those roosting within caves due to the more exposed and variable conditions present within culverts.

## Materials and methods

All methods followed ASM guidelines [43] and were approved by the Texas A&M Institutional Animal Care and Use Committee (IACUC 2015–0296).

### Study area

We conducted surveys for overwintering bats throughout Texas. Texas exhibits substantial variation in landscape, environment, latitude and climate, and is characterized by 12 level III ecoregions [44–46]. Thirty-three bat species have been documented across these ecoregions, the richest diversity of bat species found in any state in the United States [47]. We conducted surveys between November and March of 2016–2017 and 2017–2018 at 97 hibernacula, including 44 caves, 45 culverts, 3 buildings, 3 bat towers, 1 tree, and 1 bat box. We obtained access to caves through information provided by the Texas Speleological Society (TSS), Texas Cave Management Association (TCMA), Texas Grottos, Texas Parks and Wildlife biologists, and private landowners, and obtained access to other sites (i.e., buildings, bat towers, tree, and bat box) from landowners and biologists. We obtained information on historic culvert bat colonies from previous literature [48–49], from the Texas Department of Transportation (TxDOT), and from biologists. We randomly selected 77 10 x 10 km grid cells across the state using the Generalized Random Tessellation Stratified (GRTS) design [50] of the North American Bat Monitoring Program (NABat). Within those grid cells we surveyed an additional 158 culverts, of which eight contained bats. All sites included in our study were located across the following six level III ecoregions: Chihuahuan Deserts, East Central Texas Plains, Edwards Plateau, South Central Plains, Southwestern Tablelands, and Texas Blackland Prairies (Fig 1).

**Legend**
23. Arizona/New Mexico Mountains
24. Chihuahuan Deserts
25. High Plains
26. Southwestern Tablelands
27. Central Great Plains
29. Cross Timbers
30. Edwards Plateau
31. Southern Texas Plains
32. Texas Blackland Prairies
33. East Central Texas Plains
34. Western Gulf Coastal Plain
35. South Central Plains
+ Site

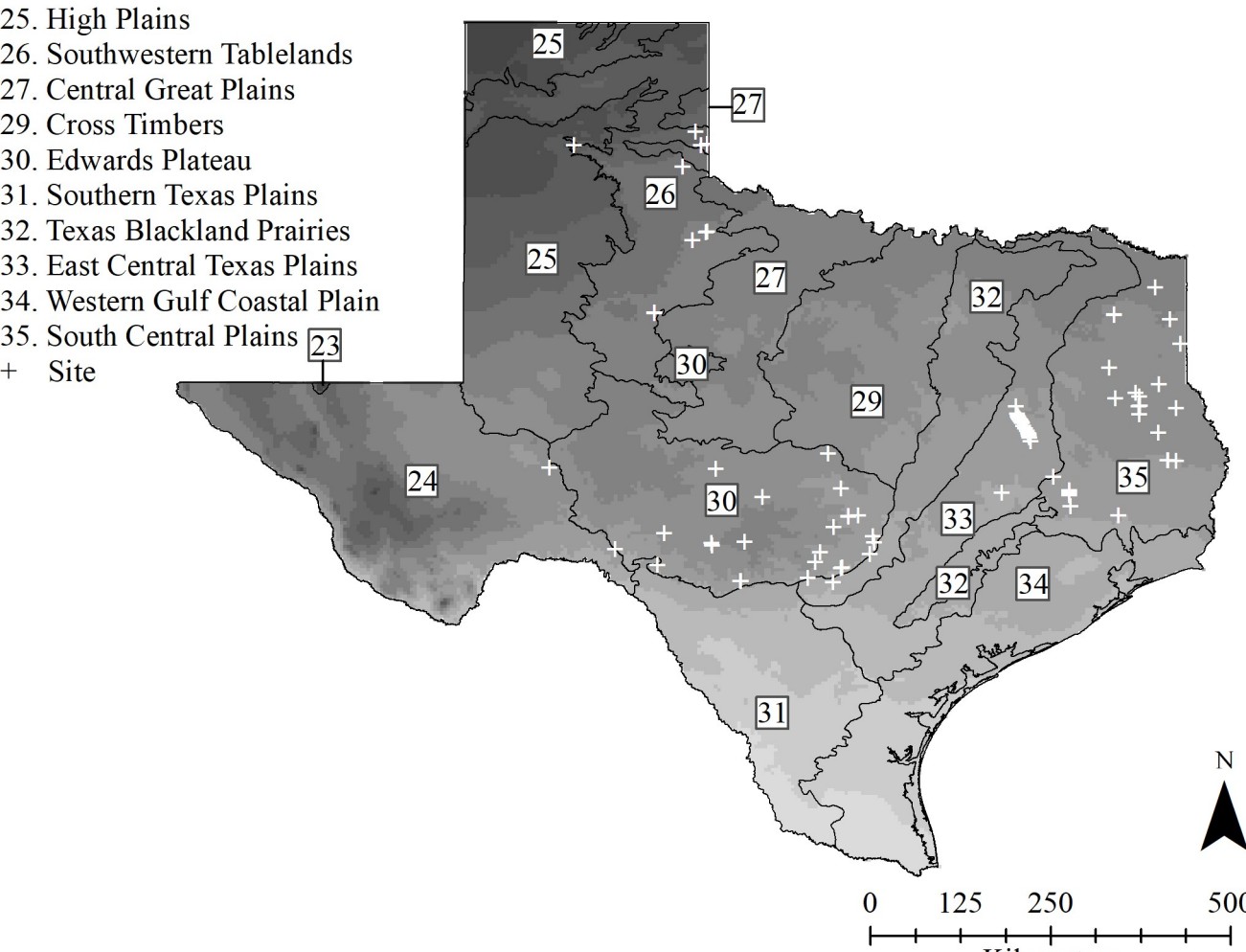

**Fig 1. Map of 97 hibernacula surveyed in Texas.** Map of the 12 level III ecoregions of Texas [46] and 30-year normal mean temperature at 4 km grid cell resolution (1981–2010) [51]. Ecoregions are numbered based on the original publication. (+) indicate the 97 hibernacula surveyed from November–March 2016–17 and 2017–18 across six of 12 level III Texas ecoregions (Chihuahuan Deserts, East Central Texas Plains, Edwards Plateau, South Central Plains, Southwestern Tablelands, Texas Blackland Prairies).

## Data collection

We conducted hibernacula surveys between 08:00–17:00 hrs. To reduce disturbance, we collected all data without any animal handling. Upon entrance into the hibernacula, we identified species present based on visual characteristics and counted the number of bats belonging to each species. We collected data on all bat species observed at each site.

To assess the relationship between external and hibernacula air temperatures, and to understand how representative our single visits to hibernacula were of the entire winter season, we deployed EL-USB-2 temperature and humidity data loggers (EasyLog, Lascar Electronics, PA, USA) programmed to record internal temperature (site temperature) every hour from October–March at two caves and two culverts occupied by bats. We retrieved average external daily

temperature from PRISM Climate Database [52] for those sites. These values were taken at 4 kilometer grid cell resolution.

To assess the microclimates within each site used by hibernating bats, we collected roost temperature and relative humidity (RH) within 5 cm of each hibernating bat or cluster using a temperature and humidity pen (Extech 445580). In order to take into account the relationship between temperature and RH, we calculated vapor pressure deficit (VPD) which describes the difference between the amount of moisture in the air and how much moisture the air can hold when saturated [53–54]. We converted RH to VPD by first calculating the saturation water vapor pressure (SWVP) for every roost temperature recorded and then multiplying SWVP by RH to produce the actual water vapor pressure (AWVP) [55]. VPD was calculated by subtracting AWVP from SWVP. We also measured skin temperatures of bats from a maximum distance of 0.3 m using a digital infrared thermometer (Extech; Waltham, MA) to gather baseline data on torpor behaviors in each species. We collected these skin temperature readings from the back of each bat. Although infrared thermometers are not the preferred method to measure bat skin temperature, they are a non-invasive and quick method of assessing temperature, and have been used in other research with success [56]. In order to ensure that we recorded reliable skin temperatures from torpid bats, we did not collect data from any bats that showed visible signs of arousal (e.g., vibrating, movement). To determine if bats were hibernating at the temperature within the hibernacula, we also determined substrate temperature adjacent to roosting bats.

## Data analysis

To test the hypothesis that site temperature is related to external temperature, we used a Pearson product-moment correlation. To test the hypothesis that Rafinesque's big-eared bats, southeastern myotis, and Mexican free-tailed bats would roost in areas of hibernacula with warmer roost temperatures and higher AWVP, and would therefore have higher torpid skin temperatures than all other species, we used Kruskal-Wallis $H$-tests (comparisons of roost temperatures and skin temperatures among species were made using separate tests). We used non-parametric statistics to analyze these data because Shapiro-Wilks tests indicated that both roost temperatures and skin temperatures for each species were not normally distributed [57]. Prior to analysis, we averaged skin temperatures, roost temperatures, substrate temperatures and VPD per site for each species to avoid pseudoreplication. We also used Pearson's product-moment correlations to test for a linear relationship between skin temperature and substrate temperature, and test for a linear relationship between skin temperature and roost temperature.

We tested our hypothesis that skin temperatures and roost temperatures of tri-colored bats hibernating in cold climates in northern Texas (Southwestern Tablelands) would be colder than those in the warm climates of southern Texas (Chihuahuan Deserts, Edwards Plateau, South Central Plains) using Kruskal-Wallis $H$-test. We performed post-hoc comparisons of groups using Dunn's tests, adjusting $P$-values using the Benjamini-Hochberg method to decrease the false discovery rate. Because we only found cave myotis in two ecoregions, we used a Wilcoxon rank-sum test to test our hypothesis that the skin temperatures of cave myotis in cold climates would be colder than the skin temperatures of cave myotis in warm climates. Finally, to compare roost temperatures and skin temperatures of tri-colored bats hibernating in culverts to those in caves between ecoregions, we used a Wilcoxon rank-sum test. All means are reported with standard deviations ($\pm$ $SD$). We considered a $P$-value $\leq 0.05$ significant for all tests. All analyses were performed in Program R v. 3.4.1 [58].

# Results

We collected data from the following seven bat species: Rafinesque's big-eared bat, Townsend's big-eared bat (*C. townsendii*), big-brown bat (*Eptesicus fuscus*), southeastern myotis, cave myotis, tri-colored bat, and Mexican free-tailed bat. Presence of these bat species varied by ecoregion [41] (Table 1).

Site temperature was highly correlated with external temperature for both occupied caves and culverts (Fig 2; $P \leq 0.05$). Between October and March, the average roost temperatures for the two caves ranged from 2.29–20.90° C (Cave 1) and 12.38–21.31° C (Cave 2) (Fig 2). During the same period, the average roost temperatures for the two culverts ranged from 0.19–23.10° C (Culvert 1) and 6.21–23.00° C (Culvert 2) (Fig 2).

Roost temperatures differed among species ($K = 30.27$, *d.f.* = 6, $P = <0.001$; Fig 3A). Rafinesque's big-eared bats were found at hibernating in areas with significantly higher roost temperatures ($\bar{X} = 22.20 \pm 4.21$) than all other species except Mexican free-tailed bats and southeastern myotis ($\bar{X} = 19.67 \pm 3.93$; $\bar{X} = 17.36 \pm 2.49$, respectively). The roost temperatures at which Mexican free-tailed bats were found did not differ significantly from the roost temperatures at which tri-colored bats, big brown bats, and cave myotis were found ($\bar{X} = 15.79 \pm 3.69$; $\bar{X} = 11.73 \pm 5.57$; $\bar{X} = 14.39 \pm 4.87$, respectively). Townsend's big-eared bats were found at significantly lower roost temperatures ($\bar{X} = 11.18 \pm 4.36$) than Rafinesque's big-eared bats, southeastern myotis, tri-colored bats, and Mexican free-tailed bats but were not significantly different to the roost temperatures at which big brown bats and cave myotis were found.

Skin temperature differed among species ($K = 41.37$, *d.f.* = 6, $P = <0.001$; Fig 3B). Rafinesque's big-eared bats and southeastern myotis had significantly higher skin temperatures ($\bar{X} = 20.65 \pm 5.85$; $\bar{X} = 18.08 \pm 2.36$, respectively) than all other species except Mexican free-tailed bats ($\bar{X} = 18.28 \pm 6.12$). The skin temperatures of Mexican free-tailed bats also did not differ significantly from tri-colored bats, big brown bats, and cave myotis ($\bar{X} = 15.07 \pm 2.87$; $\bar{X} = 10.27 \pm 3.02$; $\bar{X} = 12.41 \pm 4.05$, respectively). Townsend's big-eared bats had significantly lower skin temperatures ($\bar{X} = 9.13 \pm 3.05$) than Rafinesque's big-eared bats, southeastern myotis, tri-colored bats, and Mexican free-tailed bats, but were not significantly different to skin

**Table 1. Total count of bats by species for six level III ecoregions in Texas.**

|  | Ecoregion | | | | | |
|---|---|---|---|---|---|---|
|  | CD | ECTP | EP | SCP | ST | TBP |
| **CORA** | - | - | - | 16 | - | - |
| **COTO** | 0 | - | - | - | 52 | - |
| **EPFU** | 0 | 0 | 0 | 2 | 3 | 0 |
| **MYAU** | - | 7 | - | 13 | - | - |
| **MYVE** | 0 | 0 | 46 | - | 112 | 0 |
| **PESU** | 10 | 381 | 191 | 152 | 30 | 20 |
| **TABR** | 0 | 2 | 20 | 0 | 0 | 0 |

Total count of bats by species (*Corynorhinus rafinesquii* = CORA, *C. townsendii* = COTO, *Eptesicus fuscus* = EPFU, *Myotis austroriparius* = MYAU, *M. velifer* = MYVE, *Perimyotis subflavus* = PESU, and *Tadarida brasiliensis* = TABR) for six level III ecoregions (Chihuahuan Deserts = CD, East Central Texas Plains = ECTP, Edwards Plateau = EP, South Central Plains = SCP, Southwestern Tablelands = ST, and Texas Blackland Prairies = TBP). Data were collected from 97 hibernacula in Texas between November–March 2016–2017 and 2017–2018. (0) denotes ecoregions where the species is historically present but was not documented. (-) denotes ecoregions where the species has not been previously documented.

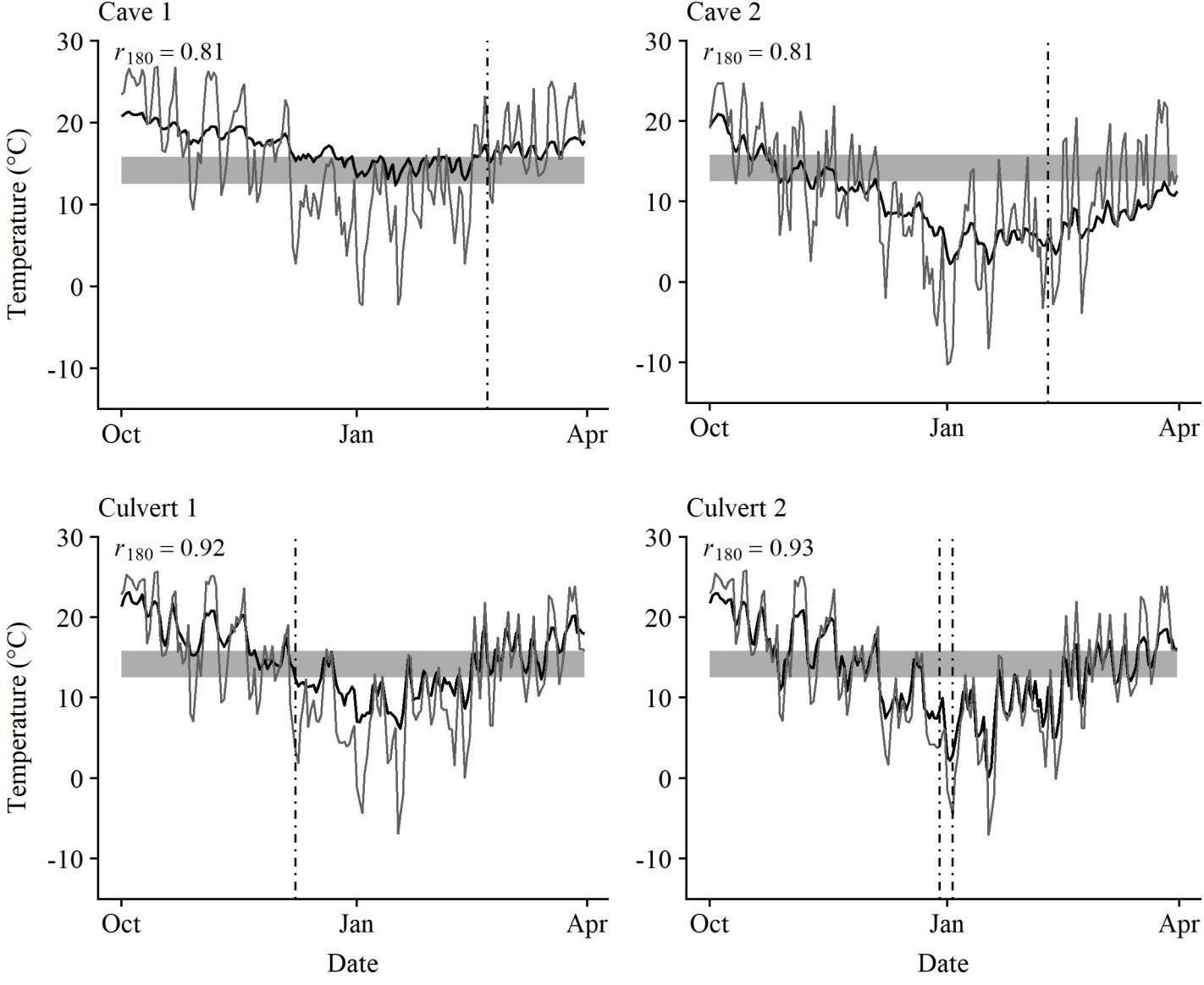

**Fig 2. Graphs of temperature profiles for cave and culvert hibernacula.** Average daily internal (black lines) and external (gray lines) temperatures (˚C) recorded at two caves and two culverts occupied by overwintering bats in Texas using an EL-USB-2 environmental data logger deployed from 1 October 2016–31 March 2017. Dashed lines indicate the dates when single-visit surveys were conducted to collect skin and microclimate temperature data from hibernating bats. Gray horizontal bars indicate the optimal growth range of P. destructans (12.5–15.8˚C) [20]. r represents the correlation coefficient between internal and external temperature with corresponding degrees of freedom.

temperatures of big brown bats and cave myotis. There was a significant positive relationship between skin temperature and substrate temperature ($r_{113} = 0.95$, $P < 0.001$). Skin temperature and roost temperature also had a significant linear relationship ($r_{102} = 0.75$, $P < 0.001$). Roost temperatures of tri-colored bats varied significantly among ecoregions ($K = 11.14$, $d.f. = 3$, $P \leq 0.01$). The roost temperatures at which tri-colored bats were found was significantly warmer in the Edwards Plateau than in the Southwestern Tablelands ($\bar{X} = 17.88 \pm 2.22$; $\bar{X} = 11.49 \pm 3.92$, respectively; $P \leq 0.01$). The roost temperatures at which tri-colored bats were found did not differ significantly between the Chihuahuan Deserts, Edwards Plateau, South Central Plains ($\bar{X} = 16.28 \pm 2.98$; $\bar{X} = 17.88 \pm 2.22$; $\bar{X} = 15.71 \pm 2.22$, respectively). Roost

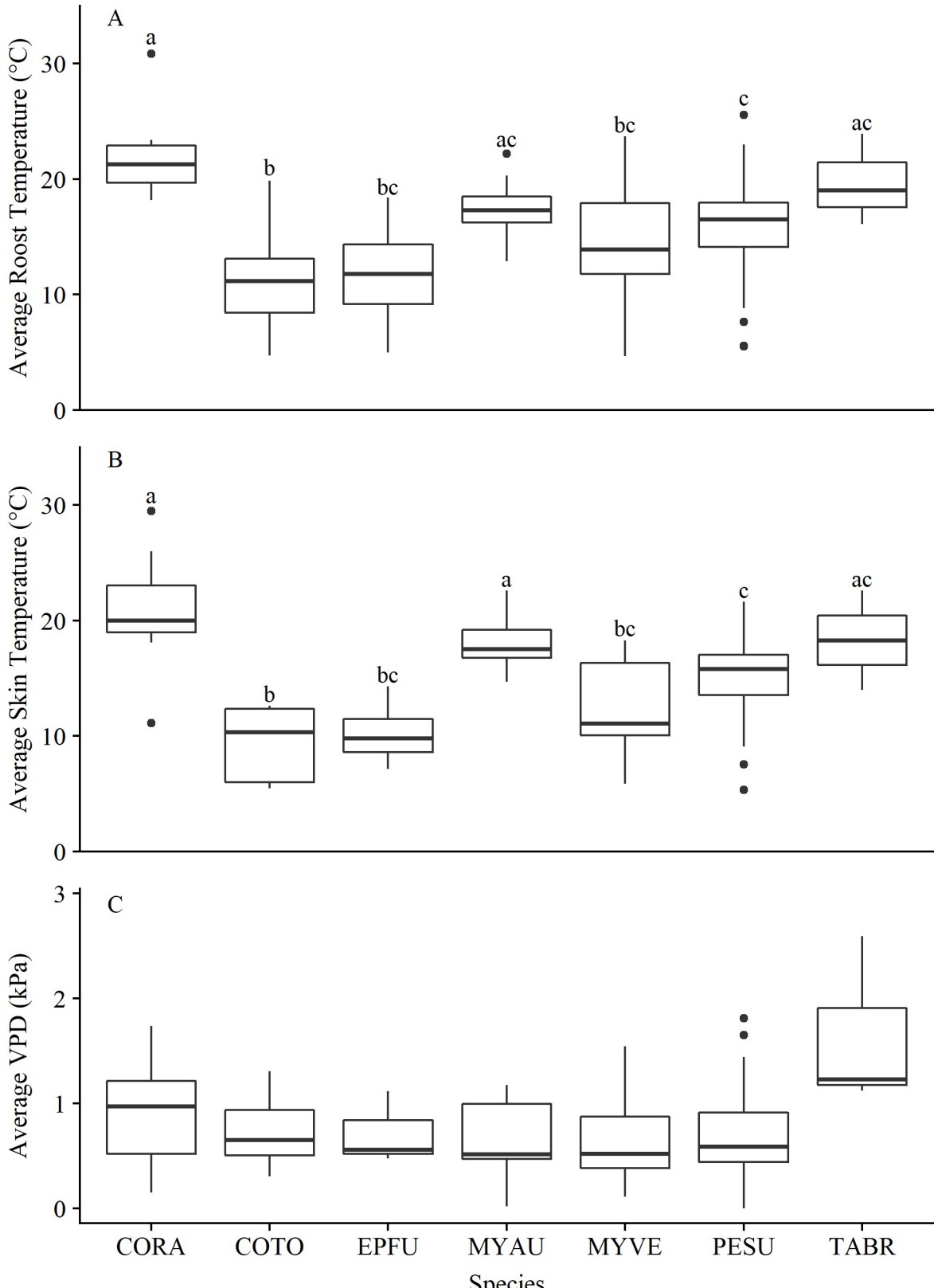

**Fig 3. Box and whisker plots of average roost temperature, average skin temperature, and vapor pressure deficit for seven bat species.** Box and whisker plots displaying the median, and upper (75%) and lower (25%) quartile range of (A) average roost temperatures,

(B) average skin temperatures, and (C) average vapor pressure deficit (VPD) collected between November–March 2016–2017 and 2017–2018 in Texas for the following seven bat species: Rafinesque's big-eared bat (*Corynorhinus rafinesquii* = CORA), Townsend's big-eared bat (*C. townsendii* = COTO), big-brown bat (*Eptesicus fuscus* = EPFU), southeastern myotis (*Myotis austroriparius* = MYAU), cave myotis (*M. velifer* = MYVE), tri-colored bat (*Perimyotis subflavus* = PESU), and Mexican free-tailed bat (*Tadarida brasiliensis* = TABR). Significant differences between groups ($P \leq 0.05$) are distinguished by letters. Filled circles indicate outliers.

temperatures of tri-colored bats hibernating in caves was not significantly different to those in culverts ($P = 0.43$; caves: $n = 24$, $\bar{X} = 15.97 \pm 3.77$; culverts: $n = 41$, $\bar{X} = 15.68 \pm 3.69$). VPD recorded adjacent to bats did not differ among species ($K = 8.86$, *d.f.* = 6, $P \leq 0.18$; Fig 3C).

Similarly, skin temperatures of tri-colored bats varied significantly among ecoregions ($K = 12.14$, *d.f.* = 3, $P \leq 0.01$). Tri-colored bats had significantly warmer skin temperatures in the Chihuahuan Deserts, Edwards Plateau, South Central Plains ($\bar{X} = 16.45 \pm 2.19$, $n = 2$; $\bar{X} = 15.34 \pm 2.77$, $n = 18$; $\bar{X} = 17.05 \pm 1.52$, $n = 2$, respectively) than in the Southwestern Tablelands ($\bar{X} = 8.73 \pm 2.46$, $n = 6$; $P \leq 0.05$). Skin temperatures of tri-colored bats hibernating in caves did not differ from those in culverts ($P = 0.19$; caves: $n = 27$, $\bar{X} = 14.32 \pm 3.73$, min = 5.35, max = 21.60; culverts: $n = 41$, $\bar{X} = 15.48 \pm 2.04$, min = 11.00, max = 18.72). Nearby VPD did not vary significantly ($K = 4.3$, *d.f.* = 3, $P > 0.05$).

The roost temperatures at which cave myotis were found was significantly colder in the Southwestern Tablelands than in the Edwards Plateau (Southwestern Tablelands: $n = 12$, $\bar{X} = 12.01 \pm 3.94$; Edwards Plateau: $n = 6$, $\bar{X} = 18.31 \pm 3.78$; $P < 0.001$). Cave myotis had significantly colder skin temperatures in the Southwestern Tablelands than in the Edwards Plateau (Southwestern Tablelands: $n = 10$, $\bar{X} = 9.50 \pm 2.12$; Edwards Plateau: $n = 6$, $\bar{X} = 16.61 \pm 1.80$; $P < 0.001$). There was no significant difference in VPD between cave myotis found in the Southwestern Tablelands and in the Edwards Plateau ($P > 0.05$).

## Discussion

We found that bats hibernate under diverse conditions across Texas, and observed broad differences in selected microclimates and torpid skin temperatures among species. Our data showed that hibernacula temperatures in Texas were suitable, and often within the optimal growth range of *P. destructans*, placing several bat species at risk of developing WNS. Although we did not investigate microclimate variability across the winter season at all sites, which constrains our ability to fully understand the available microclimates and species' susceptibility to WNS, our data add significantly to our knowledge about microclimates used by bats during winter and their suitability to the growth of *P. destructans*.

As expected, we found a strong relationship between internal and external temperatures for both caves and culverts. However, hibernacula temperatures were not simple reflections of surface temperatures and the four hibernacula that we monitored with dataloggers for the entire winter illustrate that a diversity of thermal habitats can be found in Texas. This variation can result from several factors, including aspects of local topography and the structure of the site and its entrances [59–60]. Temperatures within each hibernacula varied throughout the winter, but minimally so during the middle of the hibernation period (December–February). Thus, although it was not feasible for us to deploy microclimate dataloggers in all hibernacula, roost temperature data taken at these locations were similar to internal site temperatures indicating that our single-visit surveys captured conditions representative of the winter season. Although these temperatures are warmer than those reported for caves and mines in temperate regions, they are similar to those reported from subtropical zones [8, 11, 55, 61].

In a review of hibernacula temperatures in North America, Perry [60] found that temperatures less than 10˚ C are most suitable for hibernation, and that hibernacula in areas with

mean annual surface temperatures (MAST) greater than 10˚ C are too warm. Our data show that knowledge about hibernation ecology from the temperature region of North America may yield misleading conclusions about hibernation in the subtropical region, as we frequently found bats hibernating in temperatures greater than 15˚ C. Overall, we found that microclimates within hibernacula used by all species overlapped the temperature range of growth for *P. destructans* [20]. In fact, 22 of the 97 (22.7%) sites surveyed had microclimates within the optimal growth range (12.5–15.8˚ C) [20]. Four of the species found within these sites, the big brown bat, cave myotis, southeastern myotis, and tri-colored bat, are known to be susceptible to WNS [62]. While these data suggest high risk of WNS mortality, and *P. destructans* has been present in Texas since 2017 [31], diagnostic symptoms of WNS have yet to be documented within the state. Although WNS may yet develop in Texas, we urge researchers to investigate the possibility that hibernating bats in Texas mitigate risk to WNS through differences in winter physiology and behavior in comparison to populations farther north.

Hibernacula temperatures were not always within the optimal growth range for *P. destructans*, and in many cases temperatures less than 10˚ C were available and occupied. As we hypothesized, tri-colored bats and cave myotis in northern Texas were in deeper torpor—as indicated by lower skin temperatures—than bats of the same species in southern Texas. Tri-colored bats and cave myotis were found roosting at temperatures ranging 5.50–25.56º C and 4.67–23.67º C, respectively, and were always found torpid, as indicated by the strong correlation between skin temperature and roost temperature, and skin temperature and substrate temperature. This large range in hibernacula microclimates used by tri-colored bats is similar to values reported in previous studies (e.g., [8, 11, 48, 63–64]). The average temperature used by this species, 15.8˚ C, is at the upper end of the optimal growth range for *P. destructans*, suggesting this species is at risk in Texas. The range of microclimates used by cave myotis in our study was similar to values contained in a review by Webb and colleagues [8]. Similar to tri-colored bats, the average roost temperature for cave myotis, 14.35˚ C, suggesting these bats select microclimate conducive to growth of *P. destructans*.

Compared to tri-colored bats and cave myotis, Rafinesque's big-eared bats, Mexican free-tailed bats and southeastern myotis were consistently found in warmer areas of the hibernacula and had warmer skin temperatures, although these values did not always differ significantly from other species. This result is consistent with previous studies of Mexican free-tailed bats, which have shown this species employs shallow torpor during winter months [41, 65–66]. Unlike the Mexican free-tailed bat, Rafinesque's big-eared bat and the southeastern myotis have limited distributions within Texas, and are only found in regions of warmer climates. In particular, Rafinesque's big-eared bat is only found in the South Central Plains [41]. Compared to other bats hibernating in this region, Rafinesque's big-eared bats had the highest skin temperatures. This result is similar to that of Johnson and colleagues [42], who found this species' winter skin temperature of 13.9˚ C ± 0.6 among hibernating big-eared bats in Kentucky. Johnson and colleagues [42] also found this species frequently switched among nearby hibernacula in winter. Although we did not track movements of this or any other species, the relatively high torpid skin temperatures we observed suggests this species may also be active during winter in Texas.

Contrary to predictions, there was no difference in roost VPD between species. This indicates that—when variation in temperature is taken into consideration—the bat species studied in this manuscript do not differ in their required or preferred level of roost moisture. Across species, VPD was low, suggesting that bats were found in roosts with high levels of internal atmospheric moisture. As hibernating bats are as risk of evaporative water loss (EWL), it makes sense to find these bat species in moist environments. Although not significant, the data did suggest that Mexican free-tailed bats were found in hibernacula with slightly higher VPDs

than other species. As a species that tends to engage in shallow torpor during winter [41, 65–66], Mexican free-tailed bats are not as much at risk from EWL and thus can tolerate somewhat drier roosts.

Our work adds to previous studies documenting the use of culverts as hibernacula for tricolored bats in Texas [48–49, 67]. Culverts may act similarly to artificial caves in regions where cave-forming karst is scarce, increasing roosting options. Contrary to our prediction, tri-colored bats hibernating in culverts did not have significantly different skin temperatures than those hibernating in caves. However, the range of skin temperatures of tri-colored bats roosting within caves was larger than those in culverts. Although not reported here, our observations suggest that tri-colored bats may be selecting to roost within a given temperature range within culverts as a result of their body condition, EWL, and rates of heat loss through convection and conduction [5, 9, 68]. Raesly and Gates [69] suggested that there is an interplay between external hibernacula characteristics and internal characteristics that influence roost use. This relationship, along with the lack of caves in the region, may drive occupancy of certain culverts.

The roost temperatures, skin temperatures and actual water vapor pressures we reported herein depict microclimates selected by bats within hibernacula. Further, these data speak to the vulnerability to developing *P. destructans* infections but cannot predict WNS mortalities, as additional factors such as length of torpor bouts, winter food availability and foraging patterns must also be considered. Nevertheless, there are some important takeaways from our study. Based on our data, and provided that *P. destructans* continues to spread in Texas, we hypothesize that Mexican free-tailed bats, southeastern myotis, and Rafinesque's big-eared bats will not develop WNS in Texas. Although our data suggest that cave myotis and tri-colored bats in Texas will be affected by WNS, the lack of any WNS-affected bats in Texas to-date demands further research. Future studies should focus on gathering information on the torpor patterns (e.g., torpor bout lengths) of bats in southern US. This work will help to improve the current understanding about differences in hibernation ecology and the potential impacts of WNS on these southern occupants.

## Acknowledgments

We thank J. Carey, K. Demere, S. Goree, L. Johnston, B. Kimbell, B. Stamps, E. Whittle, and L. Wolf for their assistance in data collection. We appreciate the Texas Speleological Survey, Texas Cave Management Association, and Texas grottos for providing cave information and access. Thanks to Dr. C. Comer, J. Kennedy, Dr. W. Godwin, and D. Wright for facilitating access to private properties and data collection. We appreciate the access provided by all Texas landowners. Thanks to Texas Department of Transportation and Dr. R. Honeycutt for historic bat roost information.

## Author Contributions

**Conceptualization:** Melissa B. Meierhofer, Joseph S. Johnson, Brian L. Pierce, Jonah E. Evans, Michael L. Morrison.

**Data curation:** Melissa B. Meierhofer, Samantha J. Leivers.

**Formal analysis:** Melissa B. Meierhofer.

**Funding acquisition:** Brian L. Pierce, Jonah E. Evans, Michael L. Morrison.

**Investigation:** Melissa B. Meierhofer.

**Methodology:** Melissa B. Meierhofer, Brian L. Pierce, Michael L. Morrison.

**Resources:** Jonah E. Evans.

**Supervision:** Brian L. Pierce, Michael L. Morrison.

**Writing – original draft:** Melissa B. Meierhofer.

**Writing – review & editing:** Melissa B. Meierhofer, Joseph S. Johnson, Samantha J. Leivers, Brian L. Pierce, Jonah E. Evans, Michael L. Morrison.

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
