## [Decision Letter · Decision Letter 0]

11 Jul 2019

PONE-D-19-16454

Winter habitats of bats in Texas

PLOS ONE

Dear Ms. Meierhofer,

Thank you for submitting your manuscript to PLOS ONE. After careful consideration, we feel that it has merit but does not fully meet PLOS ONE’s publication criteria as it currently stands. Therefore, we invite you to submit a revised version of the manuscript that addresses the points raised during the review process.

We would appreciate receiving your revised manuscript by Aug 25 2019 11:59PM. To enhance the reproducibility of your results, we recommend that if applicable you deposit your laboratory protocols in protocols.io, where a protocol can be assigned its own identifier (DOI) such that it can be cited independently in the future. For instructions see: http://journals.plos.org/plosone/s/submission-guidelines#loc-laboratory-protocols

We look forward to receiving your revised manuscript.

Kind regards,

Camille Lebarbenchon

Academic Editor

PLOS ONE

Journal Requirements:

Reviewers' comments:

Reviewer's Responses to Questions

**Comments to the Author**

1. Is the manuscript technically sound, and do the data support the conclusions?

Reviewer #1: Yes

Reviewer #2: Yes

2. Has the statistical analysis been performed appropriately and rigorously? 

Reviewer #1: Yes

Reviewer #2: Yes

3. Have the authors made all data underlying the findings in their manuscript fully available?

Reviewer #1: Yes

Reviewer #2: Yes

4. Is the manuscript presented in an intelligible fashion and written in standard English?

Reviewer #1: Yes

Reviewer #2: Yes

5. Review Comments to the Author

Reviewer #1: The manuscript “Winter habitats of bats in Texas” describes hibernaculum microclimate of locations selected by bat species in the southern United States. More specifically, the authors were interested in determining the suitability of winter bat habitats in regards to white-nose syndrome. The authors determined that most of the hibernacula remained in the optimal range of fungal growth, and that skin temperatures and microclimates selected varied among species and differed from other populations within different regions. This work is especially important as the fungus that causes white-nose syndrome continues to spread into areas where we are lacking information about hibernation physiology of novel species and the microclimate conditions they chose. Data such as these will be useful in predicting the impacts of the disease as it impacts novel species.

There are a few issues that I think will improve the quality of the manuscript outlined here:

• Just a personal preference, but there are a lot of abbreviations with “T” (Tr, Ts etc). Either note the abbreviation with more detail (Tskin) or spell the temperature out.

• There were inconsistences in using abbreviations versus text throughout the manuscript. Either stick with abbreviations or text.

• It may make sense to change absolute water vapor pressure to water vapor deficit (saturation pressure – absolute pressure), which would allow the authors to compare conditions across temperature.

• Lines 77- 78: The sentence is incomplete

• Lines 114-125: These are really predictions, not hypotheses

• Line 158: change “enumerated” to “counted”

• Lines 160-165: Where within the hibernacula were the loggers placed? The distance within the hibernacula would change the temperature. Additionally, how close were the measurements from PRISM to the hibernacula? And were bats located near where the dataloggers were placed?

• Line 385: EWL has not been defined yet

• Figure 2: explain what r180 represents in the Figure caption.

• Figure 3: perhaps order by group to make visual comparisons easier

• Figure 3C: switch to water vapor deficit to make comparisons easier

Reviewer #2: This study describes the temperature of bats and their roosts during hibernation in Texas. They analyze variation between bat species, ecoregions, and roost type. The results are discussed in relation to the potential infection with the fungus P. destructans, which has not affected bats in Texas yet. Based on their temperature data, the authors make conclusions about which bat species will develop or not WNS.

I enjoyed reading this manuscript. The introduction is well written and the statistical tests particularly well described. Therefore, I only have few comments, detailed below:

1) Introduction, lines 114-118: why these 3 bat species would select higher temperatures? Please explain your hypothesis.

2) Figure 1: It might be good to have a more detailed map showing the temperatures across Texas (annual mean, or for winter only). I also suggest adding information on bat species sampled in each site to give a better idea of their range. I understand that some regions were not sampled (e.g. High Plains, Southern Texas Plains): maybe not necessary to show this information in the map.

3) In the methods, the authors explain that they counted bats during their visits. Did they test if the number of bats in the hibernacula affects (increases) the roost temperature?

4) What was the target surface measured with the digital thermometer? Did the authors observe variation of temperature across the body of torpid bats? Did they point the all bat body, or only a part (back, head, ears)? I suggest to add these details for Tskin measurements.

Minor edits:

5) Line 77 : « leading to dehydration and » ? Please rephrase.

6) Line 190 : “for each species avoid pseudoreplication”. Please rephrase.

7) Line 191 : Please correct : “used Pearson’s product-moment correlations”.

8) Line 312 : Please rephrase “…temperatures, however, and the four hibernacula…”.

9) Line 315: Please correct “Temperatures within each (?) varied…”.

10) Line 345 : Please correct “the average temperature”.

11) Line 349 : Please correct “suggests”.

6. PLOS authors have the option to publish the peer review history of their article (what does this mean?). If published, this will include your full peer review and any attached files.

Reviewer #1: No

Reviewer #2: No

---

## [Author Response · Author response to Decision Letter 0]

23 Jul 2019

Reviewer #1: The manuscript “Winter habitats of bats in Texas” describes hibernaculum microclimate of locations selected by bat species in the southern United States. More specifically, the authors were interested in determining the suitability of winter bat habitats in regards to white-nose syndrome. The authors determined that most of the hibernacula remained in the optimal range of fungal growth, and that skin temperatures and microclimates selected varied among species and differed from other populations within different regions. This work is especially important as the fungus that causes white-nose syndrome continues to spread into areas where we are lacking information about hibernation physiology of novel species and the microclimate conditions they chose. Data such as these will be useful in predicting the impacts of the disease as it impacts novel species.

There are a few issues that I think will improve the quality of the manuscript outlined here:

• Just a personal preference, but there are a lot of abbreviations with “T” (Tr, Ts etc). Either note the abbreviation with more detail (Tskin) or spell the temperature out.

• There were inconsistences in using abbreviations versus text throughout the manuscript. Either stick with abbreviations or text.

In response to both of the above comments, we have removed abbreviations in order to improve the clarity of the manuscript.

• It may make sense to change absolute water vapor pressure to water vapor deficit (saturation pressure – absolute pressure), which would allow the authors to compare conditions across temperature.

On your suggestion, we have calculated and reanalyzed our data using vapor pressure deficit. We have made the appropriate changes in text and in Figure 3C.

• Lines 77- 78: The sentence is incomplete

We have completed this sentence to “P. destructans infects dermal tissue of hibernating bats, leading to dehydration and death”

• Lines 114-125: These are really predictions, not hypotheses

We have clarified in the text where predictions or hypotheses were stated.

• Line 158: change “enumerated” to “counted”

This has been changed as suggested.

• Lines 160-165: Where within the hibernacula were the loggers placed? The distance within the hibernacula would change the temperature. Additionally, how close were the measurements from PRISM to the hibernacula? And were bats located near where the dataloggers were placed?

Dataloggers were placed within the first third of the hibernacula as- in regions where mean ambient surface temperature (MAST) is greater than 10° C, such as in Texas, bats may roost closer to the entrance where colder external air mixes with warmer air during winter (Perry, 2013). Indeed, there is some literature for the species reported in this manuscript detailing that bats tend to roost closer to the entrance of a hibernacula (big brown bat: Tuttle, 2000; tri-colored bat: Roth, 2014, Perry, 2013; southeastern myotis: Roth, 2014; Townsend’s big-eared bat: Tennessee Bat Working Group). Bats were regularly found roosting near dataloggers. While we recognize that internal temperature will vary somewhat based upon the location of the datalogger within a hibernacula, we believe that the presentation of this data provides enough evidence to suggest that our single-visit surveys captured conditions representative of the winter season for these caves and culverts. 

 In regards to PRISM, measurements were taken at 4km grid cell resolution, and this has now been clarified in the text.

Perry, R. W. (2013). A review of factors affecting cave climates for hibernating bats in temperate North America. Environmental Reviews, 21, 28-39.

Roth, Z. U. (2014). A Phenological Study of Bat Communities in Southern Mississippi Caves. Master’s Thesis, 62. https://aquila.usm.edu/masters_theses/62

Tennessee Bat Working Group. http://www.tnbwg.org/TNBWG_COTO.html. Accessed 11 July 2019.

Tuttle, M. D. (200). Where the Bats Are- Part III Cave, Cliffs and Rock Crevices. BAT magazine, 18(1), http://www.batcon.org/resources/media-education/bats-magazine/bat_article/942

• Line 385: EWL has not been defined yet

We have changed this abbreviation to read “evaporative water loss”

• Figure 2: explain what r180 represents in the Figure caption.

We have clarified that ‘r represents the correlation coefficient between internal and external temperature with corresponding degrees of freedom’.

• Figure 3: perhaps order by group to make visual comparisons easier

We not quite clear on what you are suggesting but we feel that our current figure succinctly visualizes our data and allows for comparison across variables and species.

• Figure 3C: switch to water vapor deficit to make comparisons easier

We have changed this as suggested.

Reviewer #2: This study describes the temperature of bats and their roosts during hibernation in Texas. They analyze variation between bat species, ecoregions, and roost type. The results are discussed in relation to the potential infection with the fungus P. destructans, which has not affected bats in Texas yet. Based on their temperature data, the authors make conclusions about which bat species will develop or not WNS.

I enjoyed reading this manuscript. The introduction is well written and the statistical tests particularly well described. Therefore, I only have few comments, detailed below:

1) Introduction, lines 114-118: why these 3 bat species would select higher temperatures? Please explain your hypothesis.

We have clarified that these predictions were based off the known range of the species and previous research on winter activity of the species, and have provided relevant references.

2) Figure 1: It might be good to have a more detailed map showing the temperatures across Texas (annual mean, or for winter only). I also suggest adding information on bat species sampled in each site to give a better idea of their range. I understand that some regions were not sampled (e.g. High Plains, Southern Texas Plains): maybe not necessary to show this information in the map.

We have added a gradient to the figure to show the annual mean temperatures across Texas (30 year average) taken from PRISM. Although we did not sample in some ecoregions, we have chosen to show the information on the map to provide the reader with the complete information regarding the ecoregions of Texas, and allow clear comparison to the original paper from which the ecoregions were taken (Griffith et al., 2004, 2007). In regards to the bat species sampled in each site, we believe that Table 1 provides the reader with enough information regarding the range of each species without cluttering the figure.

Griffith GE, Bryce JM, Omerick JA, Comstock AC, Rogers AC, Harrison B, et al. Ecoregions of Texas. (2 sided color poster with map, descriptive text, and photographs). 2004. [cited 29 Nov 2018]. Reston: U.S. Geological Survey. Scale 1:2,500,000.

Griffith GE, Bryce S, Omernik J, Rodgers A. Ecoregions of Texas. 27 December 2007. [cited 29 Nov 2018]. Austin: Texas Commission on Environmental Quality. Available from: ftp://ftp.epa.gov/wed/ecoregions/pubs/TXeco_Jan08_v8_Cmprsd.pdf.

3) In the methods, the authors explain that they counted bats during their visits. Did they test if the number of bats in the hibernacula affects (increases) the roost temperature?

We did not investigate if the number of bats within the hibernacula affected roost temperature. As the bats were in torpor, it is extremely unlikely that they are acting as a heat source within the hibernacula. This is validated by our strong correlation between skin temperature and substrate temperature (Line 265 of original draft). Furthermore, hibernacula were large so it is unlikely that any changes in number of bats would significantly influence roost temperature.

4) What was the target surface measured with the digital thermometer? Did the authors observe variation of temperature across the body of torpid bats? Did they point the all bat body, or only a part (back, head, ears)? I suggest to add these details for Tskin measurements.

We have now clarified in the text that we measured temperature from the bats’ backs.

Minor edits:

5) Line 77 : « leading to dehydration and » ? Please rephrase.

We have completed this sentence to “P. destructans infects dermal tissue of hibernating bats, leading to dehydration and death”

6) Line 190 : “for each species avoid pseudoreplication”. Please rephrase.

We have corrected this sentence to “…for each species to avoid pseudoreplication”.

7) Line 191 : Please correct : “used Pearson’s product-moment correlations”.

Corrected as suggested.

8) Line 312 : Please rephrase “…temperatures, however, and the four hibernacula…”.

We have rephrased the sentence to improve clarity

9) Line 315: Please correct “Temperatures within each (?) varied…”.

We have corrected this sentence to “…temperatures within each hibernacula…”.

10) Line 345 : Please correct “the average temperature”.

Corrected as suggested.

11) Line 349 : Please correct “suggests”.

Corrected to ‘suggesting’.

---

## [Editor Report · Decision Letter 1]

25 Jul 2019

Winter habitats of bats in Texas

PONE-D-19-16454R1

Dear Dr. Meierhofer,

We are pleased to inform you that your manuscript has been judged scientifically suitable for publication and will be formally accepted for publication once it complies with all outstanding technical requirements.

With kind regards,

Camille Lebarbenchon

Academic Editor

PLOS ONE

---

## [Editor Report · Acceptance letter]

1 Aug 2019

PONE-D-19-16454R1 

Winter habitats of bats in Texas 

Dear Dr. Meierhofer:

I am pleased to inform you that your manuscript has been deemed suitable for publication in PLOS ONE. Congratulations! Your manuscript is now with our production department. 

With kind regards,

on behalf of

Dr. Camille Lebarbenchon 

Academic Editor

PLOS ONE